# Transcultural Validation of a Spanish Version of the Quality of Life in Epidermolysis Bullosa Questionnaire

**DOI:** 10.3390/ijerph19127059

**Published:** 2022-06-09

**Authors:** Alvaro Rafael Villar Hernández, Fernando Molero Alonso, Álvaro Jesús Aguado Marín, Manuel Posada de la Paz

**Affiliations:** 1Dystrophic Epidermolysis Bullosa Research Association ONG DEBRA España Piel de Mariposa, 29601 Marbella, Spain; 2Department of Social and Organizational Psychology, Universidad Nacional de Educación a Distancia (UNED), 28040 Madrid, Spain; fmolero@psi.uned.es; 3School of Management, New Jersey Institute of Technology, Newark, NJ 07102, USA; alvaroaguado3@gmail.com; 4Institute of Rare Diseases Research (IIER), Instituto de Salud Carlos III, 28029 Madrid, Spain; mposada@isciii.es

**Keywords:** epidermolysis bullosa, quality of life, validation study, transcultural adaptation, Spanish QOLEB

## Abstract

Introduction: Epidermolysis bullosa (EB) is a relatively infrequent genodermatosis for which there is still no cure, and which impacts the quality of life of those that are affected by it. The Quality of Life evaluation in Epidermolysis Bullosa (QoLEB) questionnaire was specifically developed for English-speaking persons with EB. Objectives: To undertake the transcultural adaptation and analysis of the psychometric properties of a Spanish version of the QoLEB questionnaire. Method: We designed an observational study to implement the process of translation and validation of the scale in accordance with World Health Organisation guidelines. We assessed the content validity of the Spanish version with the participation of 33 adult patients who presented with four principal subtypes of EB. The subjects were examined and evaluated using the QoLEB and Short Form-36 (SF-36) questionnaires. Results: The Spanish version of the QoLEB displayed excellent internal consistency and content validity, α = 0.91. Test–retest reliability was likewise excellent (ps = 0.93), as was the reliability among subtypes (range ps = 0.82–0.93). The functional part of the QoLEB correlated well with the SF-36 physical component summary (ps = 0.70). The emotional QoLEB was moderately correlated with the SF-36 mental component summary (ps = 0.49). Significant discriminant validity existed between the global score of the questionnaire (*p* = 0.033) and the functional scale (*p* = 0.003). Conclusions: The Spanish version of the QoLEB questionnaire can be recommended for use in any subsequent studies seeking to assess the efficacy of possible treatments and care programmes in this group.

## 1. Introduction

Inherited epidermolysis bullosa (EB) is a term that encompasses a clinical and genetically heterogeneous group of low-prevalence blistering diseases, whose chief characteristic is extreme fragility of the skin and mucous membranes. This disorder gives rise to the formation of blisters in response to minimal trauma or even spontaneously [1]. It is a chronic disease for which there is currently no cure, with the result that all care and treatment received by persons affected by EB is necessarily palliative and aimed at improving the quality of life of patients and their families [2].

EB has great repercussions on quality of life, and many studies have confirmed this from a both a quantitative and qualitative perspective [3]. In 2009, the first instrument that was purpose-designed to analyse how EB affects the quality of life of those affected by it was drawn up and validated, thereby providing pertinent information for therapeutic and research purposes [4,5]. The questions in this Quality of Life evaluation in Epidermolysis Bullosa (QoLEB) questionnaire analyse the consequences of the disease on the patient by assessing the functional and emotional aspects of EB sufferers’ quality of life. This tool has since been translated into and validated in different languages and cultures [5,6,7], affording reliable comparisons and making it possible to evaluate the effectiveness of EB-specific interventions that were conducted in various non-English-speaking countries [7].

Other quality-of-life assessment instruments, both generic and dermatology-specific have also been used in a number of EB studies [3,4,5,7,8,9,10]. However, the accuracy of measurements in EB is questionable [4,7,10]. One of these instruments is the SF-36 questionnaire, a complete tool which is usually used to assess health-related quality of life globally, rather than in specific relation to a given disease [11].

## 2. Objective

This study aimed to develop a valid and culturally appropriate Spanish translation of the QoLEB questionnaire for use in Spain.

As a number of authors have noted [4,5,6,7,8,9,10,12], EB measuring instruments must be developed which can be used both in clinical practice and for research into the disease. To ensure measurement quality, it is essential that any such instrument undergoes a validation process. Furthermore, this process must not only be conducted when the instrument is drawn up in the original language but must then be repeated when the structure is altered and/or there is a change in the language in which the instrument is applied. Validating a version in a language other than the original consists of culturally adapting the questionnaire to the medium in which it is sought to be used and retesting it to ensure that it retains the appropriate psychometric characteristics to measure what it was designed to measure [13].

2020 saw the publication of the first systematic review of quality of life in EB, in which the authors conclude that the quality of life of this population must be evaluated with EB-specific instruments, such as the QoLEB questionnaire [10].

Although a Spanish version of this questionnaire has been validated in Mexico, we nevertheless felt it advisable for this questionnaire to be specifically adapted to the cultural medium in which it was sought to be used, i.e., Spain. The aim of this study was thus to validate a Spanish version of the QoLEB for Spain, study its correlation with the SF-36, and ascertain the degree of correlation of the results of this sample of Spanish patients with those that were reported by other similar papers in the literature.

## 3. Material and Methods

### 3.1. Patient and Public Involvement

This study had the authorization and support of the Association of Patients in Epidermolysis bullosa in Spain: NGO DEBRA Piel de Mariposa. The Patient Association was involved in the design and conduct of this research, including the establishment of the research question, outcomes, and the design and the execution phases of the intervention. The Association’s constant support in the dissemination of the research and the results that were obtained is noteworthy.

### 3.2. QoLEB Questionnaire

The QoLEB questionnaire evaluates two elements, namely, functional aspects (questions 1–7, 9, 10, 12, 13, 15) and emotional aspects (questions 8, 11, 14, 16, 17). For each question there are 4 response options scored from 0 to 3 points, in which a higher score denotes a worse quality of life. The questions that calculate the functional part would thus score a subtotal of 0 to 36, the emotional scale would score a subtotal of 0 to 15, and the total score for the questionnaire as a whole would range from 0 to a maximum of 51 points. The impact denoted by this global score could then be classified as follows, according to the score range achieved: very slight (0–4 points); slight (5–9 points); moderate (10–19 points); severe (20–34 points); and very severe (35–51 points) [4,7,12].

### 3.3. SF-36 Questionnaire

The SF-36 questionnaire has been translated into and validated in Spanish and assesses both positive and negative aspects of health. The final questionnaire covers 8 domains which represent the health concepts that are most frequently used in the main health questionnaires, as well as the aspects that are most closely related to the disease and its treatment. The instrument’s 36 items cover the following scales: physical function; role physical; bodily pain; general health; vitality; social function; role emotional; and mental health [14]. The 8 domains can be grouped into 2 main health components, the physical component summary (PCS) and the mental component summary (MCS) [7,14].

SF-36 scale scores of 0 to 100 have been widely used and enjoy popularity, thanks to the direct translation of their maximums and minimums to best and worst health states, respectively. Even so, the authors propose norm-based scoring for the new component summaries of the SF-36, whose principal advantage is that the results are directly interpretable with respect to the reference population [14,15]. Hence, scores based on the norm, above or below 50, respectively, indicate better or worse states of health than the mean for the reference population [14].

Our methodology, based on World Health Organisation guidelines [16] for the translation and transcultural adaptation of an EB quality-of-life assessment instrument encompasses five steps: informed consent; translation of the questionnaire from English into Spanish, followed by back translation into English again; review by a committee of experts; preliminary tests; and statistical analysis.

The methods that are applied in this validation process are similar to those described in the original paper on the validation of the Dutch QoLEB [7]. We feel that our study meets all the requirements that are demanded by the validation process of a questionnaire adapted to EB and expresses the results in a way that is straightforward and easy to understand.

### 3.4. Translation of the QoLEB into Spanish

Development of the Spanish version of the QoLEB began with its translation and intercultural adaptation by a translator whose native language was Spanish. The questionnaire was translated by a panel of experts in rare diseases and EB. The resulting version was then back-translated into English by another independent translator whose mother tongue was English, and reviewed and revised again to ensure that the QoLEB translated into Spanish conveyed the same meaning as the English original. Finally, this last revision was used to create the definitive Spanish version of the QoLEB. The Spanish QoLEB is given in Appendix A.

### 3.5. Recruitment and Design

This cross-sectional study was conducted with the collaboration of families belonging to a non-governmental organisation (NGO) known as DEBRA Piel de Mariposa, the only EB patient and family association in Spain, which currently gives support to over 300 families.

All EB-affected adult members of the NGO, Debra Piel de Mariposa, were eligible for study purposes: the inclusion criteria required patients to have Spanish as their mother tongue and an age of ≥18 years. Following the same procedure as in other similar studies [4,5,6,7], the participants were classified into 4 principal groups: EB simplex (EBS); junctional EB (JEB); dominant dystrophic EB (DDEB); and recessive dystrophic EB (RDEB). Those patients who decided to participate received two questionnaires, the QoLEB in Spanish and the SF-36, which they then completed and returned via e-mail, except for 3 cases who had to complete the questionnaires by telephone in strict compliance with the approved script. Four weeks later, the participants were asked to complete and return the Spanish-language QoLEB for a second time, in order to assess the reproducibility of the replies.

The initial number of patients who were offered the opportunity to participate was 48, but 15 of them refused or did not send the questionnaire. Of the 33 adult patients who participated in the questionnaires, 100% completed the QoLEB. Two patients failed to complete the SF-36 questionnaires and were excluded from the analysis; however, they were taken into account for the re-test of the QoLEB instrument in Spanish.

### 3.6. Analysis

A floor effect and ceiling effect was considered for individual items in cases where ≥80% of the participants obtained the highest or lowest possible scores in the first QoLEB in Spanish.

The reliability of the Spanish QoLEB was determined by an analysis of internal consistency with Cronbach’s alpha (α), based on data drawn from the first QoLEB that was completed. A value of α ≥ 0.7 was regarded as acceptable, ≥0.8 as good, and ≥0.9 as excellent [7].

Convergent validity was evaluated by comparison between the Spanish QoLEB questionnaire and the SF-36, a generic quality-of-life questionnaire that was already validated for use in Spanish [14]. This was compared against the results reported by other studies on QoLEB questionnaires that were translated into other languages and cultures, such as the Mexican and Dutch QoLEB questionnaires.

The degree of test–retest reliability was estimated by reference to the ρs (correlation value) between the general QoLEB scores registered the first and second time, respectively, with a ρs of ≥0.7 being regarded as acceptable and ≥0.8 as excellent. The level of significance was set at 5% (*p* ≤ 0.05) for all comparisons.

## 4. Results

Ascertainment of the content validity of the Spanish version of the QoLEB entailed translation of the instrument into Spanish, followed by its back translation into English, without this showing any validity problems. The results of the reliability tests displayed excellent internal consistency and construct validity, α = 0.91. Similarly, internal consistency and construct validity was good-to-excellent for the respective EB subtypes (range α = 0.83–0.93). Overall test–retest reliability was excellent (ps = 0.93), as was reliability in the subtypes (range ps = 0.82–0.93).

The breakdown showed that, of the 33 adults affected by EB who participated in the questionnaire, 22 were women (67%) and 11 were men (33%); 11 persons were diagnosed with EBS (33%), four with JEB (13%), seven with DDEB (21%), and 11 with RDEB (33%); and participants’ mean age was 38 years (range 18–83 years). Spain is divided into 17 regions or communities and the patients who have participated live in 13 of these 17 regions. (See Table 1).

The mean time for completing the questionnaire was 5.72 min, with no participant taking more than 15 min, and the questionnaire response time was considered to be short.

With respect to the SF-36 questionnaire, in terms of correlation, the results that were obtained indicated that the functional part of the QoLEB was well correlated with the SF-36 PCS (physical component summary) (ps = 0.70), and the emotional QoLEB was moderately correlated with the SF-36 MCS (ps = 0.49).

### 4.1. Assessment of the QoLEB

A confirmatory factor model was used to validate the QoLEB questionnaire in Spanish. Our analysis confirmed that the questionnaire significantly separated items pertaining to functional and emotional elements.

From these results, it could be seen that the functional and emotional variables were correlated (covariance 0.67) and that items had significant factor loadings with coefficients of 0.44 to 0.72 (see Figure 1). In addition, the results showed that there were no floor or ceiling effects, since the proportion of maximum or minimum scores in each of the questions did not exceed 70% of the subjects surveyed.

Table 2 shows the results of the Spanish version of the QoLEB with a breakdown by sex cohort and EB subtype. Women had a lower health-related quality of life than men, though these differences did not reach statistical significance. There was a statistically significant exception in the emotional section of the Spanish QoLEB (*p* = 0.007).

In general terms, the EBS and DDEB subtypes displayed a better health-related quality of life on most of the scales than the RDEB and JEB groups. These differences attained discriminant validity for the functional scale (*p* = 0.003) and global score (*p =* 0.033), but not for the emotional scale. Among the different groups there were significant differences in the functional section between the EBS-RDEB (*p =* 0.008) and DDEB-RDEB groups (*p =* 0.020), and also in the total scores between the EBS and RDEB groups (*p =* 0.085).

Table 3 shows the level of impact on the participants’ quality of life by the type of disease affecting them: 35% of interviewees reported a severe-to-very severe impact on their quality of life, as opposed to 26% who reported experiencing a slight-to-very slight impact; the remaining 39% reported a moderate impact. Similarities were observed with regard to the scores that were registered for the EBS and DDEB subtypes, and some slight differences with respect to JEB and RDEB.

The results of the 17 individual QoLEB questions are listed in Table 4.

While 100% of patients interviewed experienced pain, 18% of these reported experiencing constant pain; in terms of relationships with family and friends, EB affected 42% of participants, though in most of these cases it affected them slightly; 51% had some difficulty in eating, and the same percentage needed some type of aid for bathing or showering; and 39% experienced some limitation in writing, with this limitation being especially important in those affected by RDEB, among whom 27% were unable to write. All persons interviewed reported some limitation when it came to engaging in sports, with 81% having to avoid all or some sports; 61% reported experiencing limitations in the home, though these were slight in the great majority of cases; this percentage rose to 70% when movements took place outside the house, where scores were observed to rise with respect to those obtained in the question that analysed how EB affected movements within and around the home. Practically the same percentage was obtained, specifically 72%, when it came to shopping, and this same figure was obtained in response to the question about how EB affected the family’s financial status. EB caused 82% of the patients that were interviewed to feel frustrated, 51% to feel embarrassed, 76% to feel worried or anxious, 57% to feel depressed, and 63% to feel uneasy.

### 4.2. Short Form-36 Results

EB had a greater impact on health-related quality of life among women than among men, but the differences did not prove significant (Table 5).

Significant differences were found in the following domains: physical function (*p* = 0.010); role physical (*p* = 0.079); and general health (0.024). Comparing these between subgroups, statistical significance was achieved on the physical function scale between the EBS-JEB (*p* = 0.026) and EBS-RDEB groups (*p* = 0.024). Statistically significant differences were also found on the general health scale between the EBS and RDEB groups (*p* = 0.014).

As mentioned above, the results that were obtained for the functional part of the Spanish version of the QoLEB and the physical component summary of the SF-36 displayed a good correlation (ps = 0.701). The correlation between the results that were obtained for the emotional questions of the Spanish version of the QoLEB and the mental component summary of the SF-36 questionnaire were moderately correlated (ps = 0.4919).

## 5. Discussion

This study shows that the QoLEB, in its Spanish version, is a valid and reliable scale which makes it possible to measure the quality of life of persons who suffer from EB in Spain.

The internal consistency of the QoLEB in Spanish is high, α = 0.91, a result similar to that which was obtained in the validation of both the original (α = 0.931) and Dutch questionnaires (α = 0.905). Convergent validity is high, since the chi-squared test and factor loadings indicate that there is a high proportion of variance in common. Discriminant validity between the SF-36 and Spanish-version QoLEB questionnaires is high, HTMT = 0.681 (heterotrait–monotrait ratio of correlations), with the result being less than 0.85.

In general, the average scores that were obtained in the Spanish version of the QoLEB were higher than those recorded in the Dutch questionnaire, as was also the case in the scores that were reported for the original English questionnaire and in data collected on EB-affected adults in Brazil.

In our study, women with EB report a greater impact on their quality of life than men. These results are also found in the Dutch review [7]. Moreover, the same conclusions appear in two more articles listed in the systematic review of “quality of life in people with epidermolysis bullosa” that were published in 2020 [10].

A greater standard deviation appears in the study population in the responses that were obtained in the SF-36 than in those that were obtained in the Spanish QoLEB, a finding that suggests a wider variation among individuals by subtype in the first two questionnaires. The same occurs in the Dutch results. Our results also coincide with the results that were obtained in the first systematic review on quality of life and EB [10], in which the data that were obtained on the RDEB and JEB subtypes show a worse quality of life, in general terms, than in the EBS and DDEB groups.

As with the Dutch QoLEB and Brazilian-Portuguese QoLEB, in the Spanish QoLEB, significant differences among EB subtypes are seen on the functional and global scales but not on the emotional scale.

Although we had intended to compare and assess whether there might be differences between the Mexican and Spanish versions of the QoLEB questionnaires, this was ruled out by the impossibility of obtaining the Mexican QoLEB.

## 6. Limitations of the Study

It can be determined that the QoLEB questionnaire does not take into consideration the particularities of the paediatric age groups. The QoLEB questionnaire is a specific self-assessment tool for the burden in older children and adults with EB. However, there are studies [4,5,17,18] that have validated this questionnaire in children whose answers have been supported by their parents. In our study, we preferred to validate the questionnaire with the answers provided directly by the person with EB without the support of their caregivers.

Another weakness of our study refers to the small number of patients who participated, due to the low prevalence of the disease. In particular, we had a small representation of people with JEB (only four people). This could be explained by JEB representing just 5% of the total cases of EB [19].

## 7. Conclusions

The need for quality-of-life assessment instruments in Spanish for this group will enable us to obtain reliable measurements with which to plan effective actions and intervention strategies. The results that we obtained support the questionnaire’s construct validity and underscore its suitability for the task. In conclusion, the validation of this questionnaire in Spanish will help to optimise and evaluate the care that is offered to patients with EB in Spain, and this Spanish version of the QoLEB has been shown to be a valid and short instrument for the assessment of health-related quality of life among persons with epidermolysis bullosa.

## Figures and Tables

**Figure 1 ijerph-19-07059-f001:**
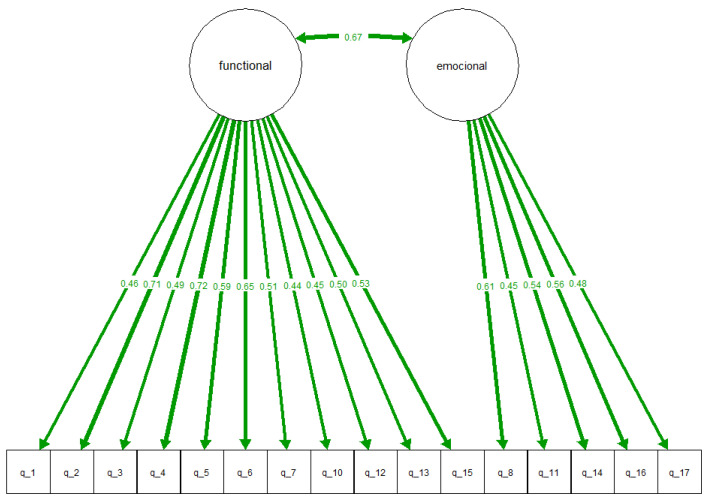
Confirmatory factor analysis of the results of the Spanish version of the QoLEB.

**Table 1 ijerph-19-07059-t001:** Sociodemographic characteristics of the participants.

Variables		EBS	EBJ	EBDD	EBDR	Total
Age	18–40	2 (6%)	3 (9%)	6 (18%)	9 (28%)	20 (61%)
	41–60	5 (15%)	1 (3%)	0	2 (6%)	8 (24%)
	+60	4 (12%)	0	1 (3%)	0	5 (15%)
	Total	11 (33%)	4 (12%)	7 (22%)	11 (33%)	33 (100%)
Gender	Male	4 (12%)	0	2 (6%)	5 (15%)	11 (33%)
	Female	7 (22%)	4 (12%)	5 (15%)	6 (18%)	22 (67%)
	Total	11 (33%)	4 (12%)	7 (22%)	11 (33%)	33 (100%)
Spanish Regions	Andalucía	0	0	4 (12%)	4 (12%)	8 (24%)
	Aragón	0	1 (3%)	0	1 (3%)	2 (6%)
	Asturias	0	1 (3%)	0	1 (3%)	2 (6%)
	Castilla La Mancha	0		2 (6%)	0	2 (6%)
	Castilla León	0	1 (3%)	0	0	1 (3%)
	Cataluña	1 (3%)	0	0	0	1(3%)
	Comunidad Valenciana	0	1 (3%)	1 (3%)	0	2 (6%)
	Extremadura	1 (3%)	0	0	1 (3%)	2 (6%)
	Galicia	0	0	0	2 (6%)	2 (6%)
	Islas Canarias	1 (3%)	0	0	0	1 (3%)
	Madrid	7 (22%)	0	0	1 (3%)	8 (25%)
	Murcia	1 (3%)	0	0	0	1 (3%)
	País Vasco	0	0	0	1 (3%)	1 (3%)
	Total	11 (33%)	4 (12%)	7 (22%)	11 (33%)	33 (100%)

**Table 2 ijerph-19-07059-t002:** Results of the Spanish version of the Quality of Life in Epidermolysis Bullosa (QoLEB) questionnaire by sex and EB subtype.

	N	Functional (0–36) Mean ± SD	Emotional (0–15) Mean ± SD	Total (0–51) Mean ± SD
All	33	11.8 ± 7.2	4.5 ± 3.1	16.3 ± 9.3
Men	11	9.8 ± 9.3	2.5 ± 1.8	12.7 ± 10.9
Women	22	12.8 ± 5.9	5.5 ± 3.1	18.2 ± 7.9
EB subtypes				
EBS	11	7.8 ± 3.9 ^†^	4.0 ± 2.8	11.8 ± 6.4 ^†^
JEB	4	16.2 ± 2.9	6.7 ± 4.5	23.0 ± 7.3
DDEB	7	7.7 ± 5.1 ^†^	4.9 ± 3.4	12.6 ± 7.7
RDEB	11	16.3 ± 8.4 ^†^	4.0 ± 2.6	20.7 ± 10.6 ^†^

^†^ Represent a statistical significant difference among EB subtypes of *p* ≤ 0.05.

**Table 3 ijerph-19-07059-t003:** Impact on quality of life by EB subtype, according to the results obtained from Spanish version of the QoLEB questionnaire.

	EBS	JEB	DDEB	RDEB	Total
Very slight (0–4 points)	1 (9%)	0	2 (29%)	0	3 (8%)
Slight (5–9 points)	4 (36%)	0	1 (13%)	1 (9%)	6 (18%)
Moderate (10–19 points)	4 (36%)	2 (50%)	2 (29%)	5 (46%)	13 (39%)
Severe (20–34 points)	2 (19%)	2 (50%)	2 (29%)	4 (36%)	10 (33%)
Very severe (35–51 points)	0	0	0	1 (9%)	1 (2%)
Total=	11 (100%)	4 (100%)	7 (100%)	11 (100%)	33 (100%)

**Table 4 ijerph-19-07059-t004:** Results of the 17 Spanish QoLEB questions, by EB subtype.

	Points	EBS n = 11 n (%)	JEB n = 4 n (%)	DDEB n = 7 n (%)	RDEB n = 11 n (%)	Total n = 33 n (%)
1. Move around the house	0	5 (45)		5 (71)	3 (27)	13 (39)
	1	6 (55)	4 (100)	2 (29)	7 (64)	19 (58)
	2					
	3				1 (9)	1 (3)
2. Bath/shower	0	9 (82)		4 (57)	3 (27)	16 (48)
	1	2 (18)	3 (75)	3 (43)	2 (18)	10 (30)
	2				1 (9)	1 (3)
	3		1 (25)		5 (46)	6 (18)
3. Pain	0					
	1	8 (73)	1 (25)	3 (43)	5 (46)	17 (51)
	2	3 (27)	3 (75)	1 (14)	3 (27)	10 (30)
	3			3 (43)	3 (27)	6 (18)
4. Write	0	9 (82)	2 (50)	5 (71)	4 (36)	20 (61)
	1	2 (18)		2 (29)		4 (12)
	2		2 (50)		4 (36)	6 (18)
	3				3 (27)	3 (9)
5. Eat	0	10 (91)		5 (71)	1 (9)	16 (48)
	1	1 (9)	4 (100)		2 (18)	7 (21)
	2				5 (46)	7 (21)
	3			2 (29)	3 (27)	3 (9)
6. Shopping	0	2 (18)		4 (57)	3 (27)	9 (27)
	1	7 (64)	1 (25)	2 (29)	3 (27)	13 (39)
	2	2 (18)	2 (50)	1 (14)	2 (18)	7 (21)
	3		1 (25)		3 (27)	4 (12)
7. Sports	0					
	1	1 (9)		3 (43)	2 (18)	6 (18)
	2	9 (82)	1 (25)	3 (43)	3 (27)	16 (48)
	3	1 (9)	3 (75)	1(14)	6 (55)	11 (33)
8. Frustration	0	1 (9)		2 (29)	3 (27)	6 (18)
	1	7 (64)	3 (75)	3 (43)	5 (45)	18 (55)
	2	3 (27)		2 (29)	3 (27)	8 (24)
	3		1 (25)			1 (3)
9. Move outside house	0	4 (36)		4 (57)	2 (18)	10 (30)
	1	3 (27)	3 (75)	3 (43)	5 (45)	14 (42)
	2	4 (36)	1 (25)		3 (27)	8 (24)
	3				1 (9)	1 (3)
10. Family	0	7 (64)	2 (50)	5 (71)	5 (45)	19 (58)
	1	4 (36)	1 (25)	2 (29)	3 (27)	10 (30)
	2		1 (25)		3 (27)	4 (12)
	3					
11. Embarrassment	0	7 (64)	2 (50)	3 (43)	4 (36)	16 (48)
	1	3 (27)		1 (14)	7 (64)	11 (33)
	2	1 (9)	1 (25)	2 (29)		4 (12)
	3		1 (25)	1 (14)		2 (6)
12.Home modifications	0	9 (82)	2 (50)	7 (100)	3 (27)	21 (64)
	1	2 (18)	2 (50)		6 (55)	10 (30)
	2				2 (18)	2 (6)
	3					
13. Friends	0	8 (73)	2 (50)	4 (57)	5 (45)	19 (58)
	1	1 (9)		3 (43)	5 (45)	9 (27)
	2	2 (18)	2 (50)		1 (9)	5 (15)
	3					
14. Anxiety	0	3 (27)		2 (29)	3 (27)	8 (24)
	1	6 (55)	3 (75)	4 (57)	6 (55)	19 (58)
	2	2 (18)	1 (25)	1 (14)	1 (9)	5 (15)
	3				1 (9)	1 (3)
15. Financial	0	5 (45)		3 (43)	1 (9)	9 (27)
	1	6 (55)	1 (25)	3(43)	5 (45)	15 (45)
	2		3 (75)	1 (14)	5 (45)	9 (27)
	3					
16. Depression	0	4 (36)		3 (43)	7 (64)	14 (42)
	1	7 (64)	3 (75)	3 (43)	3 (27)	16 (48)
	2			1 (14)	1 (9)	2 (6)
	3		1 (25)			1 (3)
17. Intimidation	0	5 (45)	1 (25)	2 (29)	4 (36)	12 (36)
	1	4 (46)	1 (25)	2 (29)	4 (36)	11 (33)
	2	1 (9)	2 (50)	3 (43)	3 (27)	9 (27)
	3	1 (9)				1 (3)

**Table 5 ijerph-19-07059-t005:** SF 36 questionnaire values, by sex and EB subtype.

	n	PF * Mean/SD	RP * Mean/SD	BP * Mean/SD	GH * Mean/SD	VT * Mean/SD	SF * Mean/SD	RE * Mean/SD	MH * Mean/SD	PCS * Mean/SD	MCS * Mean/SD
Total	31	38.0 ± 10.4	40.2 ± 8.0	38.5 ± 8.9	37.9 ± 12.7	40.3 ± 10.2	37.6 ± 13.1	42.5 ± 8.7	44.7 ± 10.5	36.6 ± 11.1	44.7 ± 10.9
Men	9	41.3 ± 11.8	42.3 ± 7.4	42.8 ± 7.0	37.3 ± 12.3	43.0 ± 9.3	41.8 ± 13.4	44.3 ± 8.3	46.4 ± 10.3	39.3 ± 10.8	46.5 ± 11.0
Women	22	36.7 ± 9.8	39.4 ± 8.2	36.7 ± 9.1	38.2 ± 13.1	39.8 ± 10.6	35.9 ± 12.9	41.8 ± 9.0	44.0 ± 10.8	35.4 ± 11.3	44.0 ± 11.0
EB subtype											
EBS	11	45.6 ± 8.2	44.9 ± 8.2	42.9 ± 5.7	46.2 ± 12.3	45.4 ± 7.6	44.9 ± 8.9	44.0 ± 6.9	46.8 ± 10.3	44.8 ± 9.0	45.7 ± 11.8
JEB	4	29.8 ± 6.9	34.8 ± 7.0	36.7 ± 10.5	37.8 ± 11.6	39.5 ± 6.9	28.4 ± 10.7	36.5 ± 8.0	42.7 ± 10.2	32.3 ± 12.3	41.1 ± 9.1
DDEB	6	36.9 ± 8.6	38.8 ± 7.4	34.1 ± 11.0	35.9 ± 13.1	33.8 ± 14.6	37.3 ± 14.5	42.7 ± 13.2	40.5 ± 13.5	34.3 ± 10.1	42.1 ± 15.5
RDEB	10	33.6 ± 10.4	38.1 ± 6.6	37.0 ± 9.2	30.0 ± 8.6	39.0 ± 9.4	34.4 ± 15.0	43.3 ± 7.9	45.9 ± 9.7	30.6 ± 9.2	46.6 ± 9.0

* PF: physical function; RP: role physical; BP: bodily pain; GH: general health; VT: vitality; SF: social function; RE: role emotional; MH: mental health; PCS: physical component summary; MSC: mental component summary.

## Data Availability

The datasets that were used and analysed during the current study are available from the corresponding author on reasonable request. Please contact the author for data requests.

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
