# Peer review of "Transcultural Validation of a Spanish Version of the Quality of Life in Epidermolysis Bullosa Questionnaire"

_ijerph, 2022, doi:10.3390/ijerph19127059_

Round 1

Reviewer 1 Report

Dear Authors, 

I read with great interest your paper. The treatment of epidermolysis is still a challenge, and patient management is based on improving the quality of life of patients and their environment. Therefore, the development of this adaptation of the questionnaire is of special interest. 

There are some points to clarify: 

1) Line 148: 33 adults patients participated. However, it should be clearly stated the initial number of patients who were offered participation and declined.

2) Age and sociodemographic characteristics of the participants in the study should be included. This is of special relevance considering that epidermolysis bullosa is a disease that usually manifests itself early in life. It should be clearly indicated at what age group (or from what age) the questionnaire is considered valid, giving the corresponding arguments. "Full-legal age" is not an appropriate definition for a scientific report, as this term may have different meanings in different geographical areas.

3) Line 289: "Epidermolysis bullosa is a low-prevalence disease which has a significant impact on 289 the quality of life of EB-sufferers and their families. ". This is not a conclusion of the present study, and therefore should be located elsewhere in the work.

4) A "limitations" section should be included, explaining which are the main limitations of the study, and the main applicability problems it could have.

Author Response

  • Line 148: 33 adults patients participated. However, it should be clearly stated the initial number of patients who were offered participation and declined.

RESPONSE: We have included the initial number of patients who were offered participation and declined.

  • Age and sociodemographic characteristics of the participants in the study should be included.

RESPONSE: We have added a table (Table I) with the age and sociodemographic characteristics of the participants in the study and also, we have added a small description of that table.

  • It should be clearly indicated at what age group (or from what age) the questionnaire is considered.

RESPONSE: Regarding the age from which this questionnaire is validated, we have changed the expression “full legal-age” (line 140) and in the new section entitled "limitations of the study" (line 297) we have explained the reasons why we have decided to include only patients ≥ 18 years of age, unlike other studies.

  • Line 289 “Epidermolysis bullosa is a low-prevalence disease which has a significant impact on the quality of life of EB sufferers and their families”. This is not a conclusion of the present study, and therefore should be located elsewhere in the work.

RESPONSE: We agree that this information should not be in this section, it would be more convenient in the "Introduction" section, and it has already been explained with other words, so we have decided to remove it. (Lines 38, 44-45)

  • “Limitations” section should be included.

RESPONSE: As recommended, we have incorporated a “Limitations” section with the main limitations of the study, and the main applicability problems (lines 297-309).

Reviewer 2 Report

The authors have aimed to validate the Quality of Life in Epidermolysis Bullosa Questionnaire translated into Spanish.

The authors have followed protocols for translation validations and correlations. The results show that the Spanish version of QoLEB can be used to evaluate the quality of life in EB patients in Spain. Considering the lack of disease-specific instruments for measuring QL in EB, this reviewer recommends that the paper is accepted and the regular use of the Spanish QoLEB is initiated in clinical practice.

Author Response

We thank to the reviewer 2 for your comments.

Reviewer 3 Report

In the case of many diseases, in order to properly assess the impact on the quality of life of patients, dedicated questionnaires should be used. This is also the case with EB, which can have a significant and multidirectional impact on patients. The authors decided to validate the Spanish-language version of the questionnaire, based on the methodology already used for the Danish version.
The results are interesting, and most importantly, the questionnaire seems to fulfill its function.
In order to prepare manuscript for the final publication, it would be worth mentioning as an appendix or in acknowledgments the group of experts who participated in the preparation of the final version of the translated questionnaire.
It should also be clarified what does not mean the possibility of comparison with the Spanish-language version was created in Mexico. What is the availability problem? Whether the validation in Mexico was published as a publication (reference not mentioned in the text).

Author Response

  • In order to prepare manuscript for the final publication, it would be worth mentioning as an appendix or in acknowledgments the group of experts who participated in the preparation of the final version of the translated questionnaire.

RESPONSE: The group of experts who participated in the preparation of the questionnaire has been mentioned in the “Acknowledgments” section (lines 321-326).

  • It should also be clarified what does not mean the possibility of comparison with the Spanish-language version was created in Mexico. What is the availability problem?

RESPONSE: In relation to the comparison with the QoLEB Spanish version from Mexico, we contacted and exchanged several emails with one of the authors, Dr. Julio Salas Alanis. However, the questionnaire was not provided, and we could not compare our version with the version of the questionnaire used in Mexico. The reference is mentioned in our manuscript “(6) Frew JW, Valdes RC, Fortuna G, Murrell DF, Alanis JS. Measuring quality of life in epidermolysis bullosa in Mexico: cross-cultural validation of the Hispanic version of the Quality of Life in Epidermolysis Bullosa questionnaire. J Am Acad Dermatol 2013;69(4):652-653” (line 351-353). The questionnaire does not appear in this publication, so we requested it but unfortunately it was never sent to us. Regularly, differences between the Spanish used in Spain and the Spanish used in other Latino America countries has to see with cultural expressions, which they do not mean the same across countries. Nevertheless, it was impossible to carry out the comparison in this case because of the lack of availability of the necessary document from Mexico.

Thank you very much for your suggestions.